



# Development of a coupled simulation framework representing the lake and river continuum of mass and energy (T-CHOIR v1.0)

Daisuke Tokuda[1], Hyungjun Kim[1], Dai Yamazaki[1], and Taikan Oki[2,3]

[1]Institute of Industrial Science, University of Tokyo, Tokyo, Japan
[2]Graduate School of Engineering, The University of Tokyo, Tokyo, Japan
[3]United Nations University, Tokyo, Japan

*Correspondence to*: Daisuke Tokuda (daisuke@rainbow.iis.u-tokyo.ac.jp)

**Abstract.** Terrestrial surface water temperature is a key variable affecting water quality and energy balance, and thermodynamics and fluid dynamics are tightly coupled in fluvial and lacustrine systems. Streamflow generally plays a role

in the horizontal redistribution of heat, and thermal exchange in lakes predominantly occurs in a vertical direction. However, numerical models simulate the water temperature for uncoupled rivers and lakes, and the linkages between them on a global scale remain unclear. In this study, we proposed an integrated modeling framework: Tightly Coupled framework for Hydrology of Open water Interactions in River–lake network (T-CHOIR). The objective is to simulate terrestrial fluvial and thermodynamics as a continuum of mass and energy in solid and liquid phases redistributed among rivers and lakes. T-

CHOIR uses high-resolution geographical information harmonized over fluvial and lacustrine networks. The results have been validated through comparison with in-situ observations and satellite-based data products, and the model sensitivity has been tested with multiple meteorological forcing datasets. It was observed that the "coupled" mode outperformed the "river-only" mode in terms of discharge and temperature in downstream of lakes; it was also observed that seasonal and interannual variation in lake water levels and temperature are also more reliable in the "coupled" mode. The inclusion of lakes in the

coupled model resulted in an increase in river temperatures during winter in mid-latitudes and a decrease in temperatures during summer in high latitudes, which reflects the role of lakes as a form of large heat storage. The river–lake coupling framework presented herein provides a basis for further elucidating the role of terrestrial surface water in Earth's energy cycle.

## 1 Introduction

The temperature of terrestrial surface water plays a vital role in biogeochemical cycles, as it affects the solubility and reactivity of materials and organismal activity (Abril et al., 2014; Ozaki et al., 2003; Webb, 1996). Water temperature can also affect water quality, resulting in adverse impacts on water availability to society. For example, the cooling efficiency of surface water used in power plants and factories is determined by water temperature, and excessively warm return flow

sometimes causes thermal pollution downstream of discharge points (Liu et al., 2020; Raptis et al., 2016; van Vliet et al.,



2016). A recent study noted that changes in surface water volume and temperature could impact the global heat budget (Vanderkelen et al., 2020). Understanding the thermodynamics of the terrestrial hydrological cycle has become increasingly important for managing freshwater environments and ecosystems, as well as developing global water policies to protect and preserve the Earth's freshwater system.

Some researchers have proposed statistical approaches to describe water temperature, such as correlating water and air temperature and the inertia of water temperature changes (e.g., Keller, 1967; Smith, 1968). Physical-based numerical models have also been developed to assess the future impacts of climate change and human activities on water temperature. As rivers and lakes are the major components of terrestrial hydrology, they are governed by extremely different dynamics, and

hence, different approaches have been adopted for each domain. Previous studies have focused on the role of rivers as horizontal transport pathways for residues from the vertical water balance between the atmosphere and the land surface (Manabe, 1969; Oki et al., 1995). A horizontal one-dimensional model for river water temperature has been developed (e.g., Sinokrot and Stefan, 1993) that assumes that sufficient mixing occurs ensuring that water temperature is uniform in a cross section (Caissie, 2006). In recent years, such a horizontally-distributed model has been applied on global scales (e.g., Beek et

al., 2012; Tokuda et al., 2019; Wanders et al., 2019) based on the development of global-scale river routing models (e.g., Yamazaki et al., 2011).

Existing studies on the thermal dynamics of lakes have mainly focused on vertical profiles, such as temperature stratification and attenuation of solar radiation (Dake and Harleman, 1969). A breakthrough in model development came through a

proposal by Henderson-Sellers (1985) for parameterization of vertical mixing to bypass explicit calculations of turbulent exchanges by shear stress. This led to the development of several numerical models (e.g., Hostetler and Bartlein, 1990) which solved a diffusion equation with the boundary conditions set at the water surface and lake bottom. These models also assumed that diffusion is primarily driven by density gradients and turbulence, allowing global-scale models to be developed (Stepanenko et al., 2013). In addition to describing the internal dynamics of lakes, lake models have been applied to

represent the lower boundary condition of atmospheric models (Dickinson et al., 1993). The impact of lakes on climate at regional scales has been widely studied since the 1990s (e.g., Hostetler et al., 1993; Small et al., 1999). Related models have also estimated that global lake evaporation will be accelerated by the changes in surface energy allocation as climate gets warmer (Wang et al., 2018).

Many modeling efforts do not treat rivers and lakes in an integrated manner. Lake models typically ignore riverine inflow and outflow and describe thermodynamics under the assumption that the elevation of the water surface is constant. For example, a previous study pointed out that the heat capacities of rivers and lakes have decreased due to lower water volumes and increased because of higher temperatures; however, these models did not properly consider the mass balance of the water budget in reality (Vanderkelen et al., 2020). Another study developed a coupled hydrodynamic and thermodynamic





model for rivers and lakes (Bonnet et al., 2000; Yigzaw et al., 2019) and demonstrated that temperature stratification in lacustrine reservoirs affects river temperatures downstream (Li et al., 2015). In particular, Yigzaw et al. (2019) used a continental-scale dataset of river networks that included lakes and reservoirs (hereinafter referred to as "river–lake network") in the United States. The river–lake networks were constructed by matching lakes with the grids on a river network dataset that previously ignored the presence of lakes and by using the relevant data on the longitude, latitude, and upstream

catchment area of each lake. However, although the upstream areas are important for water balance in lakes and reservoirs, it has been reported that this matching method does not work for some reservoirs (Shin et al., 2019).

The research reported herein initially developed a method that enabled the location and shape of lakes to be represented explicitly on a river channel network on a global scale. This technique is an extension of the upscaling method for high-

resolution topographic data, representing the shape of a hydrological unit catchment area instead of assuming a rectangular grid system (Yamazaki et al., 2009). It is possible to upscale to any required spatial resolution using this procedure. River and lake sub-models were then coupled to represent the hydrodynamics and thermodynamics of rivers and lakes on the river–lake network dataset created in this study. The modeling framework, called the Tightly Coupled framework for Hydrology of Open water Interactions in River–lake networks (T-CHOIR), conserves the mass and energy in rivers and

lakes for advection as well as the vertical heat budget. The remainder of this paper is structured as follows: Sect. 2 describes the algorithm used to develop the river–lake network dataset, Sect. 3 presents the development details of the coupling framework and the one-dimensional lake model, Sect. 4 shows the validation results of the river–lake network dataset, Sect. 5 provides the experimental configuration used to validate the framework and the corresponding results, Sect. 6 discusses the effects of thermodynamics of lakes on rivers, Sect. 7 shows the sensitivity test of the validation results of the meteorological

forcing datasets, Sect. 8 summarizes the further development of the framework, and Section 9 presents the conclusion.

## 2 Development of the river–lake network dataset

### 2.1 Harmonization of geographical information

The river–lake network was developed by upscaling high-resolution and global-scale datasets of topographical information, such as MERIT Hydro (Yamazaki et al., 2019), and lake distributions from the HydroLAKES database (Messager et al.,

2016). MERIT Hydro derives streamflow direction on a global scale using water body datasets and one of the latest elevation datasets, the MERIT DEM (Yamazaki et al., 2017). It also corrects elevation and streamlines for use in hydrological models and has a spatial resolution of 3" (~90 m at the equator). Moreover, HydroLAKES is used to identify each lake, which contains information regarding the spatial shape of lakes and reservoirs (and other attributes such as name and mean depth of each water body) of more than 1.4 million lakes and reservoirs (the term "lake" used in this manuscript includes both natural

lakes and human-made reservoirs according to the dataset). The shapefile in the HydroLAKES dataset was rasterized to the same spatial resolution as that of MERIT Hydro.





A preliminary analysis showed that the shapes of the lakes registered in HydroLAKES were often larger than the water masks in MERIT Hydro, suggesting that MERIT Hydro underestimates the area of seasonal lakes (e.g., Lake Chad) because the dataset incorporates only permanent water. Our goal for the lake model employed in this study was to represent seasonal or interannual variability of the area of various water surfaces; therefore, lake distribution data was overridden by HydroLAKES instead of using the water distribution data derived from MERIT Hydro.

When merging the datasets, we classified the lakes into two groups according to size: 1) a river–lake network that represents the connectivity of larger lakes with river channels both upstream and downstream (i.e., lakes of which area > 1 upscaled grid area), and 2) smaller lakes treated as sub-grid lakes in a numerical model (i.e., lakes of which area < 1 upscaled grid area). For example, in the case of upscaling from the original resolution (i.e., 3" to 15'), the minimum lake size in the river–lake network was 90,000 pixels. We then filled in isolated parts of each lake formed by rasterization of the shapefile (e.g., the rasterized file does not resolve a narrow conduit in Embalse Tucupido reservoir registered as 880 in HydroLAKES). We then determined the outlet of each lake using flow direction information obtained from MERIT Hydro. When inconsistencies between MERIT Hydro and HydroLAKES created multiple possible outlets for a lake, we selected the outlet with the largest upstream area, following the algorithm of HydroLAKES (Messager et al., 2016). In some cases, no outlet for a lake could be found, such as for Lake Balkhash and Large Aral Sea (registered as 12 and 13 in HydroLAKES, respectively). We found that this accurately reflected real-world geography, as both water bodies exist within closed basins. For other closed lakes such as the Caspian Sea and Lake Chad, we manually removed lake outlets which were incorrectly detected. We also adjusted the flow direction in each lake to match the outlet location and recalculated the area and number of pixels for the upstream drainage area over all grids to improve computational efficiency.

## 2.2 Upscaling method

The upscaling algorithm for the merged dataset is based on an existing method, FLOW (Yamazaki et al., 2009). FLOW divides the land area into hydrological unit-catchment distributions using a high-resolution topographic dataset (e.g., flow direction and elevation) and creates a river network with a coarser spatial resolution. The original version of FLOW did not consider lake distribution, so two additional treatments were applied to this method: we first defined lake locations within the upscaled network, with at least one upscaled grid provided for each of the larger lakes defined in the previous section to retain information such as area. When the cover fraction of the lake for each grid exceeded a threshold (80% in this study), the upscaled grid was identified as a lake; otherwise (i.e., < 80%) it was identified as a river grid. When a lake was not defined, such as one with a long and narrow shape, a single upscaled grid containing the most lake pixels was selected as the lake. The second modification to FLOW involved changing the adjustment process used for the outlets of the unit catchments. The original version of FLOW adjusted the location of unit catchment outlet to equalize the area and length of the catchment. This process was modified so that the location of the lake outlet was the same as the outlet of the unit





catchments. The river–lake inlet was moved closer to the boundary between the lake and the river. The former of the modifications is used to couple the lake model with the reservoir operational model, and the latter aims to lengthen the channel area for better calculation efficiency (i.e., increase the time step of the river model). Figure 1 shows examples of the upscaled river–lake network dataset.




**Figure 1: Example of upscaled river–lake network dataset. Figures shows results under four different configurations for Lake Biwa and Yodo River basin in Japan, (a) 15' without lakes, (b) 15' with lakes, (c) 6' without lakes, and (d) 6' with lakes.**



## 3 Model description

The river–lake network dataset provides fundamental information for use in a framework that couples river and lake models.
The network explicitly relates the models to corresponding grids and represents the horizontal connectivity between them.
Because the physical schemes used in this study to represent hydro- and thermodynamics in rivers are identical to an existing
model (Tokuda et al., 2019), this section focuses on the lake model and the coupling framework after briefly summarizing
those riverine schemes.

### 3.1 River model

The river model used in this study is HEAT-LINK (Tokuda et al., 2019), which is fully coupled with a river routing model,
CaMa-Flood (Yamazaki et al., 2011, 2013). This model solves the conservation laws of mass, momentum, and energy for
one-dimensional channels.

CaMa-Flood solves the conservation of momentum law by approximating it using a local inertial flow equation, and this
enables efficient computation (Bates et al., 2010). The equation is discretized explicitly using a forward-time central-space
scheme to simulate the time evolution of the state variables. CaMa-Flood calculates discharges in river channels and
floodplains as prognostic variables and diagnoses the cross-sectional shape (e.g., the water depth and area in river channels
and adjacent floodplain). This river routing model accounts for fluvial dynamics in river channels and the floodplain with
objective parameterization based on high-resolution topographic data. The introduction of floodplain inundation in addition
to channel storage tends to reduce changes in water level and leads to improved reproducibility of seasonal discharge
variability in large continental rivers, such as the Amazon (Yamazaki et al., 2011).

HEAT-LINK solves water and ice mass and energy conservation laws by calculating heat fluxes, including short- and
longwave radiation, sensible heat, latent heat, and frictional heat (Tokuda et al., 2019). The methods used to calculate fluxes
are identical to those used in existing studies (Kondo, 1992; Hondzo and Stefan, 1994; Webb and Zhang, 1997). As HEAT-
LINK considers varying water surface area and decaying absorption of downward shortwave to water depth, it can represent
the role of flood inundation in water temperature changes. By considering the effects of riverine hydrodynamics on
thermodynamics, the model properly produces the variability for water temperature not only in ice-free and ice-covered
channels, but also in regions with bi-modal seasonality of water temperature (Tokuda et al., 2019).

### 3.2 Lake model

A simple one-dimensional lake model was implemented to represent the water budget and thermodynamics in lakes. To
reproduce the exchanges between rivers and lakes, the lake model conserves mass and energy while considering horizontal
advection in addition to the vertical exchange. All input data, except the meteorological forcing data, were provided by



HydroLAKES and the Global Reservoir and Dam Database (GRanD) (Lehner et al., 2011), as described in the following
sections.

### 3.2.1 Hydrodynamics

The lake model used in this study represents seasonal variation in water depth and discharge above a lake bottom elevation
derived from HydroLAKES, because the water surface elevation is a downstream boundary condition, and the outflow is an
upstream boundary condition for the river model. The water balance in each lake is expressed by the following equation:

$$\frac{\mathrm{d}S}{\mathrm{d}t} = P - E + Q_\mathrm{in} - Q_\mathrm{out} = A(p - e) + Q_\mathrm{in} - Q_\mathrm{out} \tag{1}$$

where $S$ (m³) is the water storage in each lake; $t$ (s) is the time, $P$ and $E$ (m³ s⁻¹) are precipitation and evaporation,
respectively; $Q_\mathrm{in}$ and $Q_\mathrm{out}$ (m³ s⁻¹) are the inflow from and outflow out of the lake, respectively; $A$ (m²) is the lake surface
area, and $p$ and $e$ (m s⁻¹) are the precipitation and evaporative loss per area, respectively. $Q_\mathrm{in}$ and $Q_\mathrm{out}$ also consider
backflow at both the lake inlet and outlet. While the precipitation per area is given as input data and the river discharge at the
inlet from rivers is calculated by the river model, the outflow and evaporation are calculated from a weir formula and the
thermodynamics model in the following section.

This study set geomorphological boundary conditions by estimating the depth–area relationship according to the Global
Reservoir Geometry Database (ReGeom) (Yigzaw et al., 2018). In that study, the water surface of each reservoir was
extracted from satellite images, and the depth–area relationship was estimated to match the total storage of the reservoir
registered in GRanD. However, the dataset assumed several shapes for horizontal and vertical cross-sections, and the
estimation of the surface area and volume of water may have led to large inconsistencies with the reported values for some
reservoirs. Therefore, in this study, the vertical shape assumed in the ReGeom was generalized to derive a new depth–area
relationship that is consistent with the surface area and volume of water derived from other datasets. In this respect, the area
attenuation rate $f(r)$ was calculated by Eq. (2) when $r = z/D_0$, where $z$ is the vertical distance from the origin at the
elevation wherein the water surface area is at its maximum, and $D$ is the total depth from the origin to the bottom of the
reservoir.

$$f(r) = \begin{cases} (1 - r^2)(1 - r)^a & \left(V_0 < \tfrac{2}{3} A_0 D_0\right) \\ 1 - r^a & \left(\tfrac{2}{3} A_0 D_0 \le V_0 < A_0 D_0\right) \\ 1 & (A_0 D_0 \le V_0) \end{cases} \tag{2}$$

where $A_0$ (m²) and $V_0$ (m³) are the surface area and volume of water, respectively, when the water depth is $D_0$. $a$ is the shape
scaling parameter. $D_0$, $A_0$, and $V_0$ are input from GRanD, and $a$ is derived as follows:

When $V_0 < 2/3\,(A_0 D_0)$,





$$p = \frac{V_0}{A_0 D_0} = \frac{a+4}{(a+2)(a+3)} \quad \therefore a = \frac{-5p + 1 + \sqrt{p^2 + 6p + 1}}{2p} \tag{3}$$

When $2/3\,(A_0 D_0) \leq V_0 < A_0 D_0$,

$$p = \frac{V_0}{A_0 D_0} = 1 - \frac{1}{a+1} \quad \therefore a = \frac{1}{1-p} - 1 \tag{4}$$


otherwise (i.e., $A_0 D_0 \leq V_0$), $D_0$ was updated as $V_0/A_0$, and a constant depth–area relationship was assumed. The area was also assumed to be constant with respect to water depth for lakes not registered in GRanD.

As the spatial shape of the lakes was obtained from HydroLAKES, the depth–area relationship was also estimated only for
those lakes that exhibited consistent volumes in HydroLAKES (attribute name is Vol_total) and GRanD (attribute name is Cap_mcm). For example, this condition excluded Lake Baikal (HydroLAKES ID is 11) where the volumes registered in the two databases were 23,615,000 and 46,000 mcm, respectively.

This study assumes that the outlet of each lake is a rectangular cross section, and applies the weir formula to estimate the
outflow $Q_{\text{out}}$ as follows:

$$Q_{\text{out}} = {}^{2}\!/_{3} \sqrt{2g}\, C_d B h^{3/2} \tag{5}$$

where $g$ (m s$^{-2}$) is the gravitational acceleration, $C_d$ is a correction coefficient, $B$ (m) is the width of the outlet, and $h$ (m) is the height from the top of the weir to the water surface. A similar formula is used in the existing global model, WaterGAP Global Hydrology Model (WGHM) (Döll et al., 2003; Meigh et al., 1999), but it considers the relationship using a parameter
known as "active storage." Equation (2) assumes a situation in which subcritical flow requires a specific water depth at the overflow section; it then transitions to supercritical flow and forms a free-falling water cascade. In this study, based on the above considerations, the amount of outflow from each lake was calculated in the following three ways:

$$Q_{\text{out}} = \begin{cases} kBh_0^{3/2} & (h_1 \leq {}^{2}\!/_{3}\, h_0) \\ kB(h_0 - h_1)^{3/2} & ({}^{2}\!/_{3}\, h_0 < h_1 \leq h_0) \\ -kB(h_1 - h_0)^{3/2} & (h_0 < h_1) \end{cases} \tag{6}$$

where $k$ is a correction coefficient (to be set 5.0 because $C_d \approx 1.7$), and $h_0$ and $h_1$ (m) are the height of the lake surface and
downstream surface from the weir height, respectively. When a lake flows into a river channel, this study assumes that the river water depth in each unit catchment is uniform and gives $h_1$ as the downstream water depth; otherwise, $h_1$ is calculated from the surface elevation of the downstream lake or the ocean. Additionally, environmental flow is represented using a simplified method as follows: The minima among 1) 20% of inflow to lake and 2) discharge to maintain the water depth of the river grid located immediately downstream greater than 0.5 m is considered to be the environmental flow, and is taken as
the outflow when discharge based on the extended weir formula is smaller.





$Q_{\text{out}}$ is supposed to be zero in an inland lake with no outlet. However, preliminary results showed that several inland lakes (e.g., Small Aral Sea, HydroLAKES ID is 130) were identified where the water level did not reach equilibrium and continued to increase even after a spinning-up calculation due to two factors: 1) overestimation of riverine inflow caused by

the lack of knowledge about some processes including groundwater infiltration and water withdrawals; 2) absence of negative feedback due to the unavailability of the water depth–area relationship in drier regions, where $P - E$ is negative. In a basin with such a lake, the water level of the surrounding river increases along with the lake due to backflow, resulting in unrealistic ranges (this backflow continues until the inflow to the lake is balanced by the vertical water balance of the lake). In this study, to stabilize the water level of such inland lakes within realistic range, we calculated the outflow using the

method described below and discharged it out of the system. This treatment is identical to that of the river model that does not represent lake dynamics.

### 3.2.2 Thermodynamics

The lake thermodynamics model implemented in this study is a vertical one-dimensional model in which the water temperature is calculated for each vertical layer, and therefore the horizontal distribution within the lake is not resolved. This

model is also able to represent the phase change between water and ice. The vertical structure (i.e., maximum number of vertical layers and thickness of each layer) is configurable, and the number of active layers and thickness of the bottom layer vary along with lake depth which is defined in order from the water surface to the lake bottom. In the case of a lake that has an area with varying water level as described in Section 3.2.1, the volume of each layer is calculated with consideration given to this relationship.


The heat budget of water is expressed using Eq. (7) (Hostetler and Bartlein, 1990):

$$\frac{\partial T}{\partial t} = \frac{1}{A}\frac{\partial}{\partial z}\left(AK\frac{\partial T}{\partial z}\right) + \frac{1}{c_w \rho_w A}\frac{\partial (A\phi)}{\partial z} \tag{7}$$

where $T$ (°C) is the water temperature, $z$ (m) is the depth, $A$ (m$^2$) is the horizontal area of a lake, $K$ (m$^2$ s$^{-1}$) is eddy diffusivity, $c_w$ (J kg$^{-1}$ °C$^{-1}$) is the specific heat capacity of water, $\rho_w$ (kg m$^{-3}$) is the density of water, and $\phi$ (W m$^{-2}$) is the shortwave

radiation. To calculate $K$, this study uses the method of Henderson-Sellers (1985), which considers wind-driven diffusion and buoyant convection, and the exponential attenuation of shortwave radiation is used to calculate $\phi$ (Dake and Harleman, 1969).

The boundary conditions are the heat flux exchanges at the water surface and lake bottom. The heat fluxes at the water

surface include up- and downward, short- and longwave, sensible heat, and latent heat. The methods for calculating these fluxes are identical to those of the river model. Assuming a well-mixed condition, the surface temperature of the water is assumed to be the same as the first (uppermost) layer and not defined as skin temperature. This calculation is only performed when the surface is not covered by ice completely. For the ice-water mixed case, the boundary condition for the surface flux





is the weighted mean of the ice-free and ice-covered areas; the boundary condition beneath the ice is the conductive flux

between the water and ice. The heat flux from the lake bottom is assumed to be zero, in accordance with existing models

(Goudsmit et al., 2002; Hostetler and Bartlein, 1990; Joehnk and Umlauf, 2001).

Many lakes in mid- and high latitude regions experience ice formation during the winter, and several lake ice models have

been developed in past decades (Gu and Stefan, 1990; Hostetler, 1991; Hostetler et al., 1993; Croley and Assel, 1994;

Patterson et al., 1998) The representation of the temporal evolution of ice volume in this study is consistent with such models

based on the heat budget of the ice. Additionally, this study uses a simplified version of the ice shape parameterization from

the Great Lakes Advanced Hydrologic Prediction System (AHPS), a watershed model developed for the Great Lakes region

by the Great Lakes Environmental Research Laboratory (GLERL) (Croley and Assel, 1994),

The boundary conditions for the heat budget of the ice cover are that the ice temperature adjacent to the atmosphere $T_{si}$ (°C)

and to water $T_{bi}$ (°C) are equal to the atmospheric temperature $T_a$ (°C) and melting point of water $T_m$ (= 0 °C), respectively,

$$\begin{cases} T_{si} = \min(T_a, T_m) \\ \quad T_{bi} = T_m \end{cases} \tag{8}$$

The heat balance at the ice surface is expressed as follows:

$$\phi_{dli} - \phi_{uli}(T_{si}) - \phi_{Hi}(T_{si}) - \phi_{ci} = 0 \tag{9}$$

where $\phi_{dli}$, $\phi_{uli}$, $\phi_{Hi}$, and $\phi_{ci}$ (W m⁻²) are the downward longwave, upward longwave, sensible, and conductive heat fluxes

at the ice surface, respectively.

The temporal change in the ice volume is expressed as follows:

$$-\rho_i \gamma_i \frac{dS_i}{dt} = A_i(\phi_{ci} - \phi_{iw} + \phi_{dsi}) \tag{10}$$

where $\rho_i$ (kg m⁻³) is the ice density, $\gamma$ is the fusion heat (= 333,500 J kg⁻¹), $S_i$ (m³) is the ice volume, $A_i$ (m²) is the ice area,

and $\phi_{iw}$ and $\phi_{dsi}$ (W m⁻²) are the conductive heat flux from ice to water and the absorption of shortwave radiation by the ice

body, respectively. $\phi_{iw}$ is calculated using the same method as that of an existing river temperature model (Beek et al., 2012).

Our model also assumes that ice loss due to sublimation is negligible according to the GLERL AHPS model (Croley and

Assel, 1994).


While a two-dimensional horizontal model computes the ice fraction in each grid in one lake (Goyette et al., 2000), a one-

dimensional model such as the T-CHOIR needs to consider the ice fraction at a the sub-lake scale. The GLERL AHPS

computes the time evolution of ice thickness and area using heat exchange with the atmosphere, water, snowfall, and

evaporation. Our study simplified this representation by adding two assumptions: the ice shape change is only caused by

exchanging heat with the atmosphere; the thickness of the ice is only determined by area. Under these assumptions, ice thickness is proportional to the square root of the ice area.

The heat budget of water and ice is solved as follows. First, the heat flux from the ice surface to the ice body is calculated by solving the heat balance at the ice surface. With this and the other two fluxes, an increase or decrease in the mass of the ice is

calculated. If the ice melts completely in a timestep, the heat flux from the ice to the water is recalculated. If there is no ice in a lake, these processes are skipped. The model then computes the heat flux from the water surface to the water body by solving the heat balance at the water surface. The heat fluxes from the ice and the water surface are combined as an upper boundary condition for the water body. The amount of incoming shortwave radiation into the water body is also weighted by the area of ice and water. The heat budget of each layer of the water body is solved using an implicit scheme with a

staggered grid. In the layers below the melting point, the amount of ice formation is calculated and added to the surface ice. Water from the melted ice is added to the water surface layer.

Because the river model does not solve vertical distributions of mass and heat fluxes within the channel, both rainfall and the water inflowing to a lake are added to the surface layer. The snowfall and ice inflows are also added to the ice cover. The

model assumes that the temperature of precipitation is equal to the air temperature, with the minimum temperature of rainfall and maximum temperature of snowfall set to the melting point of water. The model decreases the water volume from the surface to lower layers onward for outflow and evaporative loss. The model reanalyzes the ice shape, the layer structures, and the water temperature profile by mixing the temperature of the existing layers from top to bottom, thereby conserving the mass and energy of the water.

**3.3 Implementation of coupling interface**

**3.3.1 Grid system**

T-CHOIR reads the dataset of a two-dimensional river–lake network derived by the upscaling method described in Sect. 2.2, and then vectorizes it to a one-dimensional array for better computational efficiency. The river grids are arranged in the following order: channel grids from rivers to rivers, lake inlet grids from rivers to lakes, and river mouth grids to the ocean.

Following the flow direction, the lake outlet is connected to a river, another lake, or ocean. As the lake outlets and inlets are matched to the outlets of the unit catchments, the discharge at the lake outlets and inlets are calculated using the lake and river model, respectively. These discharges can be negative, which indicates backflow (e.g., a negative value at a lake outlet means that the net water flow is from downstream into the lake).



### 3.3.2 Data exchanges and communications between model components

As the river and lake models were developed separately, a wrapper interface was required to share interdependent boundary conditions between the sub-models. The T-CHOIR framework was built using a coupler that encapsulates the sub-models and ensures topological consistency in time and geography for their communications. Figure 2 shows the sequence of the temporal integrations and data exchange in the coupler. In this study, corrections to the river and lake outflows were made to ensure that the outflow is smaller than the storage discharge. Data exchange between the models occurred prior to correction,

which is necessary for some conditions (e.g., backwater).





**Figure 2: Flowchart of the calculation of T-Choir. The solid (dashed) arrows indicate the order of the calculations (data exchange). The arrow branches indicate that each process can be computed in parallel, but the data exchange takes place at the same time.**

The coupler also shares common information for both models, such as time steps and meteorological forcing data. The time step is determined such that the river model satisfies the Courant–Friedrichs–Lewy (CFL) condition, and it is used for the lake model as well.

Our framework structure has two significant advantages: 1) encapsulation of stand-alone sub-models, which allows them to be developed or easily replaced, and 2) provision of a convenient testbed to turn each sub-model on or off. In the following section, we compare the results of three experiments to distinguish the interactions between rivers and lakes: "coupled," "river-only," and "lake-only."

## 4 Validation of harmonized geographical information

This section shows the validation results of the harmonized geographical information by comparing the upstream area with the survey-reported value in GRanD for some reservoirs, instead of the upstream area based on the hydrography information in HydroLAKES. Although the spatial resolution of the following validations of numerical simulations is 15', the validation in this section was conducted in 6' resolution to have a higher number of lakes available for validation (only 14 reservoirs are resolved at a 15' resolution). The upscaling method implemented in our study conserves the upstream area of the high-
resolution input data for all grids; therefore, the upstream areas were identical for lakes resolved between the 6 and 15-arcmin networks.

Figure 3 shows a comparison between the calculated values obtained in this study and the reported values in GRanD for the upstream area of each reservoir. The correlation coefficient was 0.852 for all 103 matched reservoirs. The calculated values
of only eight reservoirs were greater than 200% or less than 50% of the reported values. Those eight reservoirs are further compared to hydrological data gathered by the United States Geological Survey (USGS) and Australian National Committee on Large Dams (ANCOLD) (Table 1). In the rest of this section, we discuss three reservoirs that did not agree with the upstream area, both in our data and in other data sources.





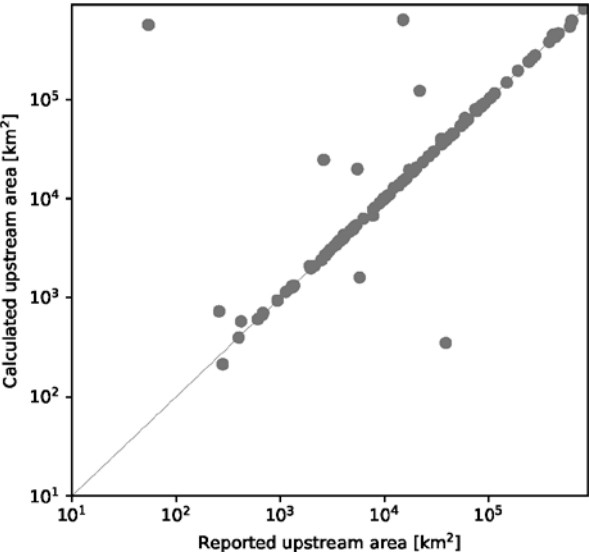


**Figure 3: Comparison between calculated upstream area (km²) in the river–lake network developed in this study (vertical) and reported values registered in GRanD (horizontal).**

**Table 1: List of reservoirs whose areas (km²) differ by more than a factor of 2 or 0.5 between calculated and reported upstream values. The ID column provides the lake's identifier in HydroLAKES, and the columns showing the values in literature and the reference refer to upstream areas described in other literature from GRanD.**

| ID | Name | Calculated value (km²) | Reported value in GRanD (km²) | Value in literature (km²) | Reference |
|---|---|---|---|---|---|
| 764 | Lake Sharpe | 638023.6 | 15126.0 | 645684.0 | USGS 06442700 |
| 815 | Lake Eufaula | 122613.4 | 21769.0 | 123081.4 | USGS 07244800 |
| 830 | Lake Moultrie | 350.3 | 38850.0 | 38331.8 | USGS 02172002 |
| 835 | Cedar Creek Reservoir | 24599.4 | 2608.0 | 2608.1 | USGS 08063010 |
| 1062 | Iovskoye Reservoir | 19900.2 | 5510.0 | 5488.8 | HydroLAKES |
| 1293 | Lake Iron gates | 566946.1 | 54.0 | 560682.4 | HydroLAKES |
| 1701 | Lake Pedder | 730.1 | 258.0 | 733.0 | ANCOLD |
| 8978 | Lake Cascade | 1596.8 | 5776.0 | 1592.8 | USGS 13244500 |

Lake Moultrie is located at the uppermost point of the Cooper River Basin located in the United States. The Cooper River has a length of 230 km, and the larger Santee River flows through the northern part of the basin. Lake Marion
(HydroLAKES ID 828) is located at the confluence of the Wateree River and the Congaree River, which are tributaries of the Santee River. There are two outlets in Lake Marion: one to the Santee River and the other to Lake Moultrie. However, the network developed in this study assumes that each lake has only one outlet according to HydroLAKES, which means that

such diversions are not represented. In fact, the reported and calculated values of the upstream area of Lake Marion are almost identical (38,073.0 and 38,060.2 km$^2$, respectively), while the reported upstream area of Lake Moultrie is more

similar to that of Lake Marion. This implies that the reported value of Lake Moultrie includes Lake Marion and its upstream area.

The Cedar Creek Reservoir is located within a sub-basin of the Trinity River, and the lake discharge flows into the main stem of the river. However, in the river–lake network dataset developed here, the reservoir is located on the mainstem. This

problem is related to flow direction information in MERIT Hydro, and this issue has already been reported to the data developer.

The Iovskoye (Иовское) reservoir is in the Kovda (Ковда) River Basin. Given the basin map and the fact that the entire Kovda River has a basin area of 25,600 km$^2$ (O'Sullivan and Reynolds, 2008), it is considered that the estimate of the

upstream area of 19,900.2 km$^2$ in this study is reasonable.

## 5 Validation of integrated simulation framework

### 5.1 Simulation configuration

The spatial resolution of the models was set to 15', and the river–lake network dataset was upscaled to the same resolution. As a result, 369 lakes from around the world were represented at the target resolution. Their area and volume accounted for

51% and 92% of the total area and volume of all lakes in HydroLAKES, respectively. As described in Sect. 2 and 3, the geomorphic information (e.g., mean depth, mean surface area, and sill height) were obtained from the HydroLAKES and GRanD datasets. The depth–area relationships were defined for 79 out of 369 lakes, and the area of the other lakes was set to be constant.

Three different meteorological forcing datasets were prepared to investigate the uncertainty caused by the datasets: GSWP3 (Kim, 2017), JRA55-ELSE (Kim, 2020), and Prcp-GPCCLW90 (Kim et al., 2009). The associated spatio-temporal resolutions are shown in Table 2. To produce the amount and temperature of runoff, a land surface model, MATSIRO (Takata et al., 2003), was employed by using each meteorological forcing dataset. These outputs have the same spatial resolution as the input data (i.e., 0.5 degrees, 1 degree, and 1 degree, respectively), and the temporal resolution is one day.

The main results were based on the GSWP3 forced simulation, and we discuss the sensitivity of the results in Sect. 5.1.



**Table 2: Summary of meteorological forcing data used in this study.**

| Product name | Spatial resolution (deg) | Temporal resolution (h) | Reference |
|---|---|---|---|
| GSWP3 | 0.5 | 6 | Kim (2017b) |
| JRA55-ELSE | 1.0 | 3 | Kim (2017a) |
| Prcp-GPCCLW90 | 1.0 | 6 | Kim et al. (2009) |

This study assumed constant values for several physical parameters of rivers and lakes. The albedo and attenuation rate for
the shortwave radiation of water (ice) were assumed as 0.1 (0.6) and 0.1 (10.0) m$^{-1}$, respectively, and the absorption rate of
the shortwave radiation at the water surface was set to 0.4. Future research will examine the variations in the parameters by
considering other processes including solar zenith angle and water turbidity. Previous studies have proposed empirical
equations to calculate outflow for the Great Lakes (Croley and Assel, 1994), which we have adopted. The vertical structure
is configured as 5 layers of 0.2 m, 6 layers of 0.5 m, 8 layers of 2 m, 4 layers of 5 m, 2 layers of 10 m, and 2 layers of 20 m.


The calculation period used in this study was 2000 to 2002, and the spin-up was carried out by repeating the simulation in
2000 twenty times. We manually set initial values for water depth and temperature profile for several lakes to remove the
interannual variability in the lake water depth and the bottom temperature after the spin-up. To observe interactions between
rivers and lakes, we conducted two separate experiments, in addition to the "coupled" experiment. The first used a
conventional river model that did not include lakes (the "river-only" experiment). The river network dataset used in this
experiment was slightly different from that used with the "coupled" experiment due to modification of the outlets of unit
catchments to the inlet and outlet of lakes. As this river network was created from the same dataset (MERIT Hydro) using
the same algorithm, the comparison should be reasonable. In the second simulation, we turned off the river model and
computed only lakes (the "lake-only" experiment). The simulation ignored the direct flow from one lake into another in
order to exclude the effects of riverine dynamics on lakes.

### 5.2 Reference data

The framework was validated by comparison with in-situ and satellite observation data for the following five variables: 1)
river discharge, 2) river temperature, 3) lake surface elevation, 4) lake surface temperature, and 5) vertical profiles of lake
temperature. The Pearson correlation coefficient (CORR), bias (BIAS), and root-mean-square difference (RMSD) were used
in evaluations. Additionally, for river discharge, normalized BIAS and RMSD calculated by the mean observed value
(hereafter pBIAS and pRMSD) were used because of the wide range of absolute values.

In-situ observation data of river discharge and temperature were collected by the Global Runoff Data Centre (GRDC) and
the Global Environmental Monitoring System (GEMS). Validation was performed at a monthly time scale. Stations with

data longer than a year were selected, which resulted in 148 and 75 of GRDC and GEMS reference sites, respectively,
located downstream of lakes. Stations located upstream were excluded because the backwater effect was negligible (Fig. S1).

To validate the seasonal change in water surface elevation of the lake, the G-REALM dataset (Birkett et al., 2018) was used.
This dataset provides the water surface elevation of lakes with areas greater than 100 km$^2$ based on satellite altimetry remote
sensing. We used the EGM96 referenced data, which is identical to the MERIT DEM. To identify a lake between
HydroLAKES and G-REALM datasets, latitude and longitude information were used. The consistency of the matched pairs
was then manually checked. As a result, 318 out of the 340 lakes in the G-REALM dataset were matched, and 152 lakes
were resolved in the river–lake dataset. This validation was performed for 132 lakes that have data longer than a year.

For lake surface temperature, the GloboLakes (Carrea and Merchant, 2019) dataset was used. It provides multiple satellite-
based estimations at 0.05-degree global grids on a daily scale for 979 lakes. The observation lake grids were arithmetically
averaged to compare with the lake model in this study, which does not represent the horizontal distributions; only quality
flags of 4 (acceptable quality) and 5 (best quality) were taken. This dataset was matched to HydroLAKES using the same
method as above, which resulted in 878 matching lakes, of which 200 were resolved in the river–lake network. The
validation was performed on 124 lakes that have data longer than a year.

The vertical profile of lake water temperature for lakes in North America was validated against The Water Quality Portal
(WQP). WQP covers lakes globally, but vertical temperature profiles were only available for the North American region.
The comparison between the simulated data and observations is instantaneous because the vertical observations are made
only a few times a year. Up to three observation locations were selected for each lake.

**5.3 River discharge at downstream areas of lakes**

Figure 4 summarizes the reproducibility of river discharge simulated by the T-CHOIR framework downstream of the lakes
for the "coupled" and "river-only" simulations. Overall, it showed an improved performance when the lake was considered
(i.e., "coupled"). However, some rivers showed the limited impact of the lakes (e.g., the Lena and Amazon rivers). The
impacts were mainly found in two aspects: 1) reduction in the overestimation of discharge and 2) dampening of the
amplitude of the seasonal variations in the "river-only" simulation. For example, at Cornwall station in the St. Lawrence
River (Figure 5 (a)), located downstream of the Great Lakes, discharge from the "coupled" experiment shows better
performance than that of the "river-only" simulation due to higher evaporative loss at the lake surface, which affects the
basin-wide water balance significantly. The reduced seasonality is evident at the Volgograd station of the Volga River,
Manitou Rapids station of the Rainy River, and Above Kazan Falls station of the Kazan River (Figure 5 (b), (c), and (d)). In
these cases, the incorporation of lakes leads to the dampening of peak discharge, because the lake plays a role as a buffer
between flux and storage.



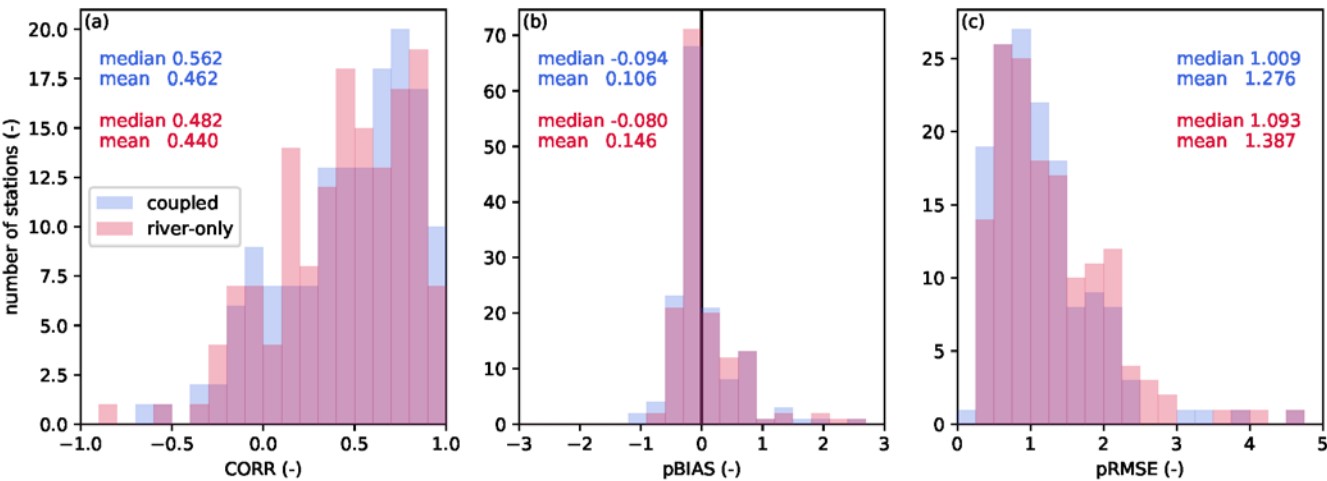

**Figure 4: Comparison between reproducibility indices of river discharge: bars are the histograms of each index, and the numbers indicate the associated median and mean values. Blue (red) bars and written values show the results of the "coupled" ("river-only") simulation. (a) CORR, (b) pBIAS, and (c) pRMSD.**

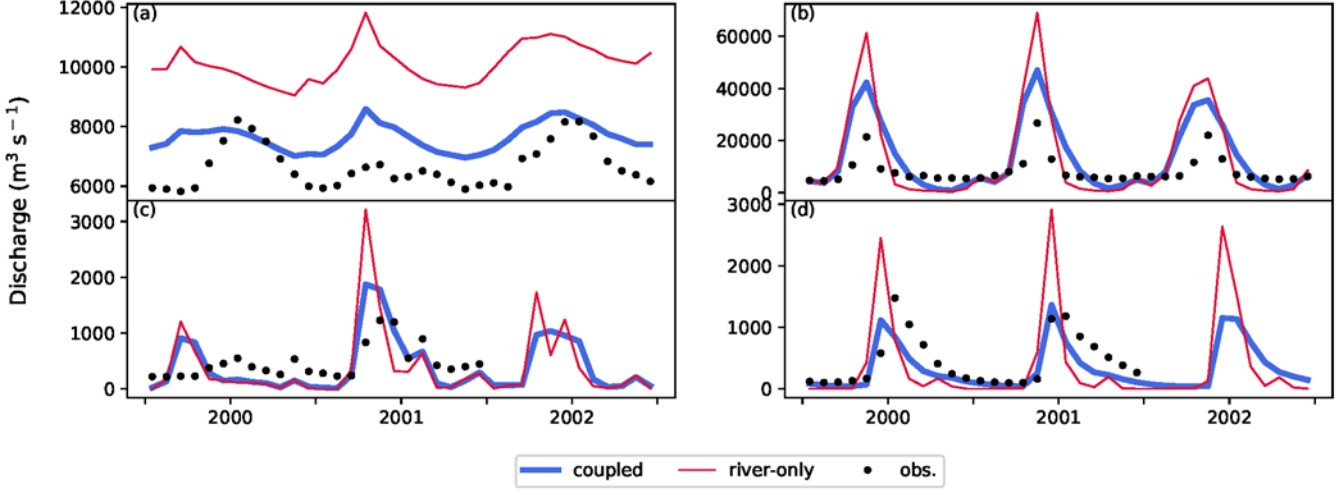

**Figure 5: Timeseries of monthly mean values of river discharge (m$^3$ s$^{-1}$). Black dots show the observed values, and blue (red) line shows values simulated by the "coupled" ("river-only") model. (a) Cornwall station in the St. Lawrence River, (b) Volgograd station in the Volga River, (c) Manitou rapids station in the Rainy River, and (d) Above Kazan falls station in the Kazan River. Station codes of GRDC are 4143550, 6977100, 4213211, and 4214090, respectively.**

## 5.4 River temperature at downstream of lakes

The effect of lakes on river temperature is summarized in Figure 6. It is evident that all performance metrics improved when lakes were presented. Positive or negative river temperature biases were reduced significantly. In particular, for a number of



stations in Brazil, coupling of river and lake models reduced the overestimation of water temperature. For example, improvement at 00MS13SM2000 station in Rio Santa Maria (Figure 7 (a)) was due to an increase in heat release resulting from an increase in residence time. At Hamilton traffic bridge station in the Waikato River and Puerto Libertad station in the
Parana River, the simulations were improved (Figure 7 (b) and (c), respectively) due to the fact that warmer water near the surface flows out of lakes due to thermal stratification, among which improvement observed at Puerto Libertad station is significant during the cold season. On the other hand, the incorporation of the lake model led to a lower performance for some Russian stations, such as the Neva River and Cheboskarskoye Reservoir stations (station code RUS00014 and RUS00029, respectively). The former is located downstream of Lake Ladoga (HydroLAKES ID is 10), just before it flows
into the Gulf of Finland (Figure 7 (d)). Lake Ladoga is a large lake that spans more than $1.5°$ north to south. Our framework was unable to capture the temperature peak, especially in summer. We speculate that inflow carrying warmer water from the southern upstream area and the missing representation of sub-lake-scale dynamics may be the cause of such shortcomings and suggest selecting a river scheme for lakes where horizontal flow predominates in addition to vertical mixing. In this respect, a previous study proposed a method for calculating water temperature in lakes using a river model that considers a
lake to be a wider river (Beek et al., 2012). A similar shortcoming was found in the Gorkovsky Reservoir (HydroLAKES ID 109).

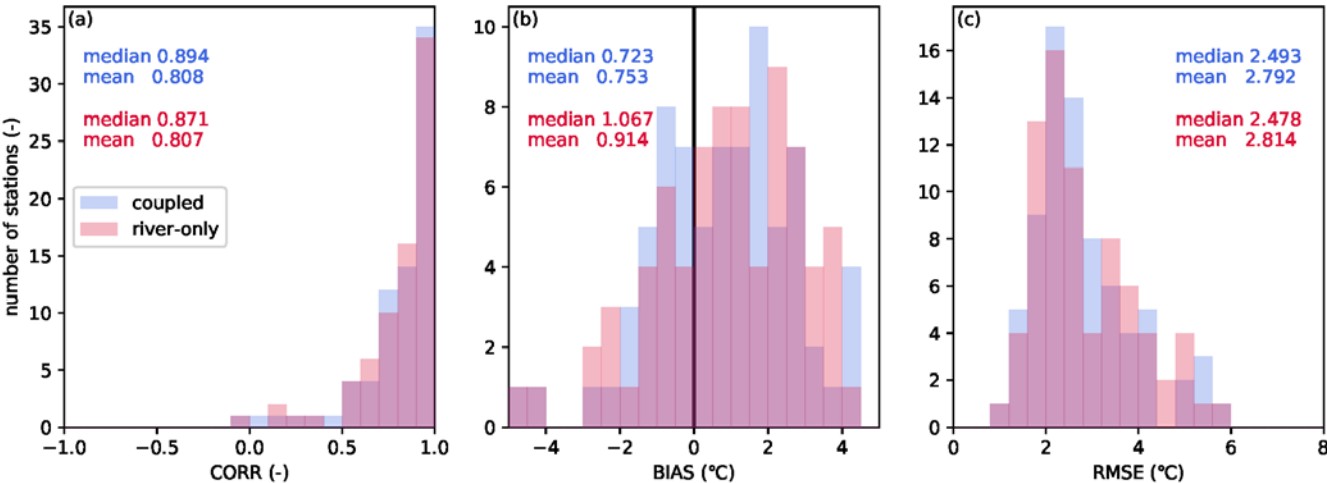

**Figure 6: Comparison between reproducibility indices of river temperature. Bars show the histogram of each index, and the**
**numbers indicate associated median and mean values. Blue (red) bars and written values show the result of "coupled" ("river-only") simulation. (a) CORR, (b) BIAS (°C), and (c) RMSD (°C).**

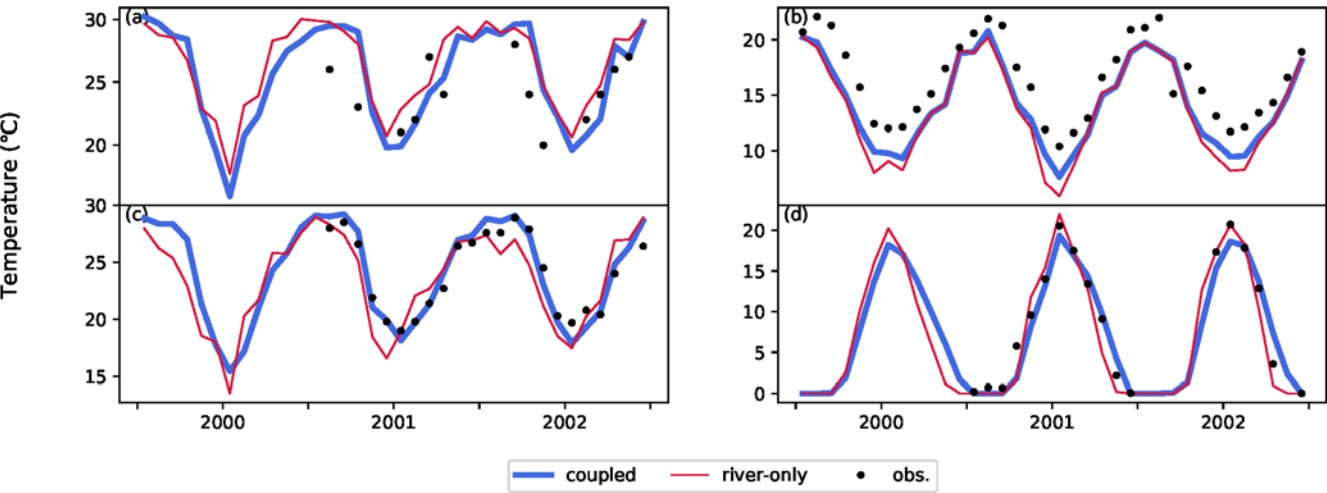

**Figure 7: Timeseries of monthly mean values of river temperature (°C). Black dots show the observed values, and blue (red) line shows values simulated by the "coupled" ("river-only") model. (a) 00MS13SM2000 station in Rio Santa Maria, (b) Hamilton traffic bridge station in the Waikato River, (c) Puerto Libertad station in the Parana River, and (d) Neva River station in the Neva River. Station codes in GEMS are BRA01900, NZL00013, ARG00001, and RUS00014, respectively.**

## 5.5 Lake water surface elevation

The following sections compare the results from the "coupled" and "lake-only" experiments. A comparison of the performance metrics (i.e., CORR, BIAS, and RMSD) for the water surface elevation in the 132 lakes is shown in Figure 8. We noted that the "lake-only" simulation did not reach the equilibrium state even after the 20-year spin-up, showing a steady increase or decrease (Figure 9). This is not surprising, given the imbalance between precipitation and evaporation. Therefore, the "lake-only" simulation is validated only for CORR in Figure 8. The typical examples are shown in Figure 9 (a) and (b), which show the time series data for Lake Superior and Champlain. The water surface elevations of those lakes keep increasing in the "lake-only" simulation because of the mass imbalance; precipitation is greater than evaporation, which is consistent with observations (Bennett, 1978; Smeltzer and Quinn, 1996). However, the incorporation of riverine dynamics allows for variation in lake water level within a reasonable range, as the river inflow and outflow play a role in dampening the water level change in lakes. According to Figure 8 (a), although the seasonality of the lake surface elevations is dominated by the water budget within the lake (i.e., precipitation minus evaporation), the topographic information surrounding the lakes plays a crucial role in reproducing the absolute value of the water surface elevation in addition to the water budget in between rivers (i.e., in and outflow). Therefore, applying the coupling framework is potentially beneficial for a long-term Earth system simulation. The "coupled" simulation also reproduces the range of seasonal variability in Lake Superior. However, the water level in Lake Champlain tends to be overestimated during the wet season. A tuned empirical



equation gives the outflow from Lake Superior. At the same time, the model possibly underestimates negative feedback

between water level and outflow in Lake Champlain.

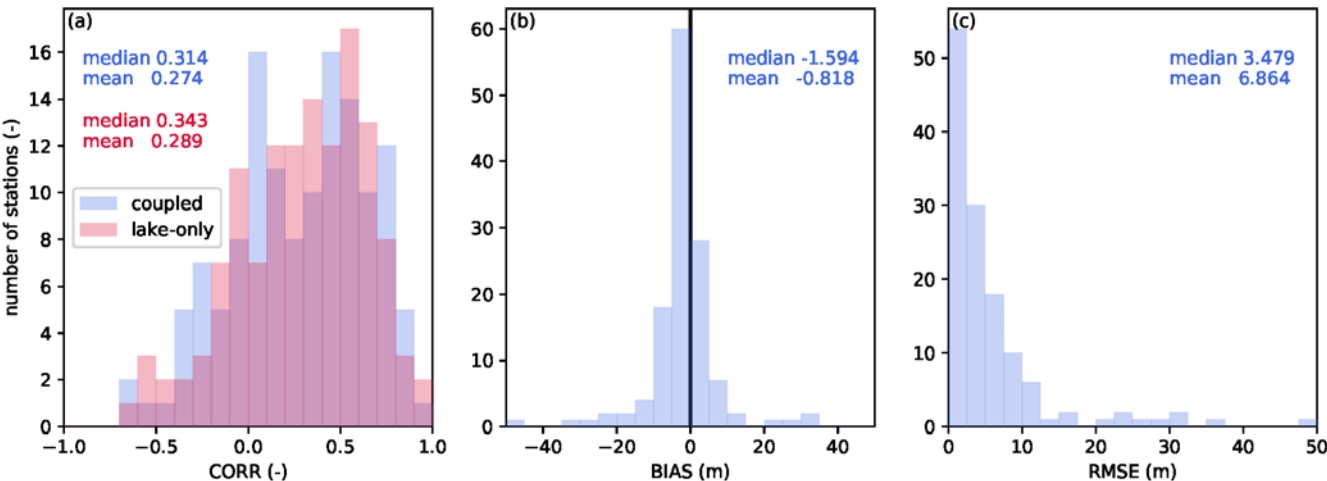

**Figure 8: Comparison between reproducibility indices of lake surface elevation. Bars are the histogram of each index, and written**
**values show the associated median and mean values. Blue (red) bars and writing show the results of the "coupled" ("lake-only")**
**simulation. (a) CORR, (b) BIAS (m), and (c) RMSD (m).**

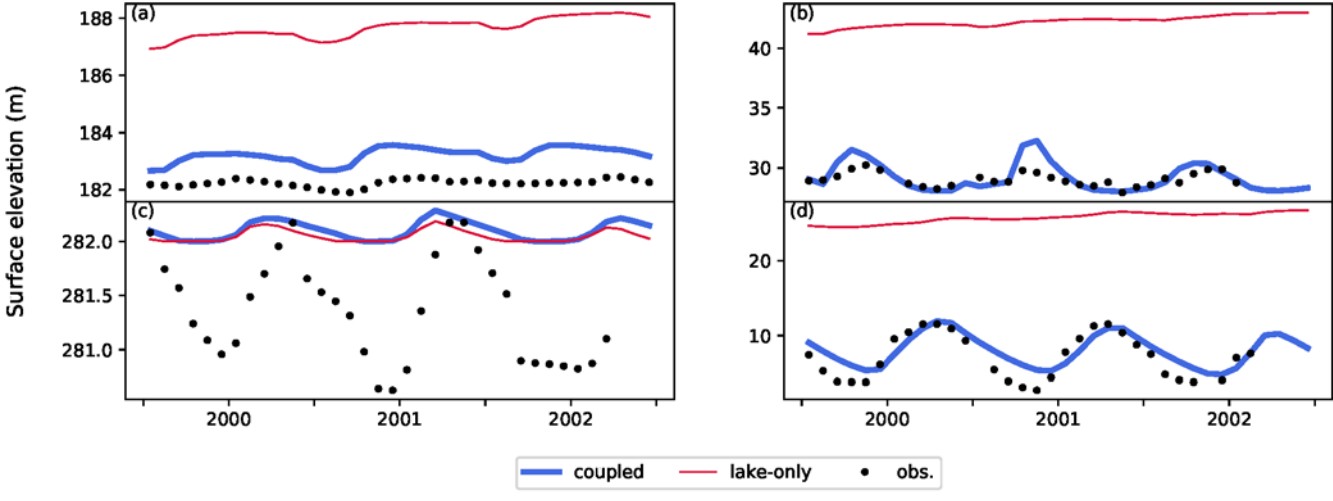

**Figure 9: Timeseries of monthly mean values of lake surface elevation (°C). Black dots show observed values, and blue (red) line**
**shows value simulated by the "coupled" ("lake-only") simulation. (a) Lake Superior, (b) Champlain, (c) Chad, and (d) Tonle Sap**
**(HydroLAKES IDs are 5, 64, 15, and 153, respectively).**



For Lake Chad (Figure 9 (c)), both the "coupled" and "lake-only" experiments overestimated the water level. The water level

was reproduced relatively well during the wet season, but there was a considerable discrepancy between observations. This is because the simulated water level cannot be lower than the given bottom elevation (282 m). To reproduce lake water levels accurately, it is important to treat this lake in two or three parts separately (Gao et al., 2011; Lemoalle et al., 2012). Such topography-induced impacts on lake surface extent and level are rather significant in dry regions. Therefore, a possible solution is to integrate sub-lakes defined based on precise topography information via the T-CHOIR framework.


However, the "coupled" simulation shows high applicability for use in Lake Tonle Sap (Figure 9 (d)). This lake joins the Mekong River in the downstream area, which functions as a natural floodwater reservoir due to backflow from the river during the wet season. Although our framework cannot be compared directly with the water budget of a previous estimation (Kummu et al., 2014) because the river model in this study does not distinguish tributary flows and overland flow, the

seasonal change in the simulated outflow from the lake indicates a similar pattern (in which the outflow is negative for several months, mainly from July to September). It is of note that we do not consider temporal changes in the lake area fixed at 2415.98 km$^2$, which is within the minimum range of this lake (Kummu et al., 2014).

### 5.6 Lake surface temperature

Between the "coupled" and "lake-only" experiments, the difference in the water temperature estimates was not significant,

but the performance measured by the metrics showed slight improvements in the "coupled" simulation (Figure 10). The improvement in bias metric is relatively apparent. Although the lake water storage and heat capacity are significantly larger than the fluvial and thermal inflow, the riverine impacts on the lake temperature were found during the summer (Figure 11). The temperature difference between the lakes and associated inflows is mainly due to the lower shortwave absorption rate per unit volume in the lakes. Shortwave radiation reaches deeper lake water after attenuation, but shallower river water can

effectively absorb radiation. Most of the variation in water temperature near the surface is explained by heat exchange at the interface with the atmosphere. However, it is also influenced by lake depth, as shown in an intercomparison experiment using multiple vertical one-dimensional lake temperature models (Stepanenko et al., 2013). It implies that two factors may cause differences in water temperature between the "coupled" and "lake-only" simulations, such as: 1) the effect of improved lake depth estimation and 2) the effect of differences in water temperature between rivers and lakes. For the latter

factor, further comparative experiments are required with a lake model that resolves the spatial distributions of water temperature within a lake.





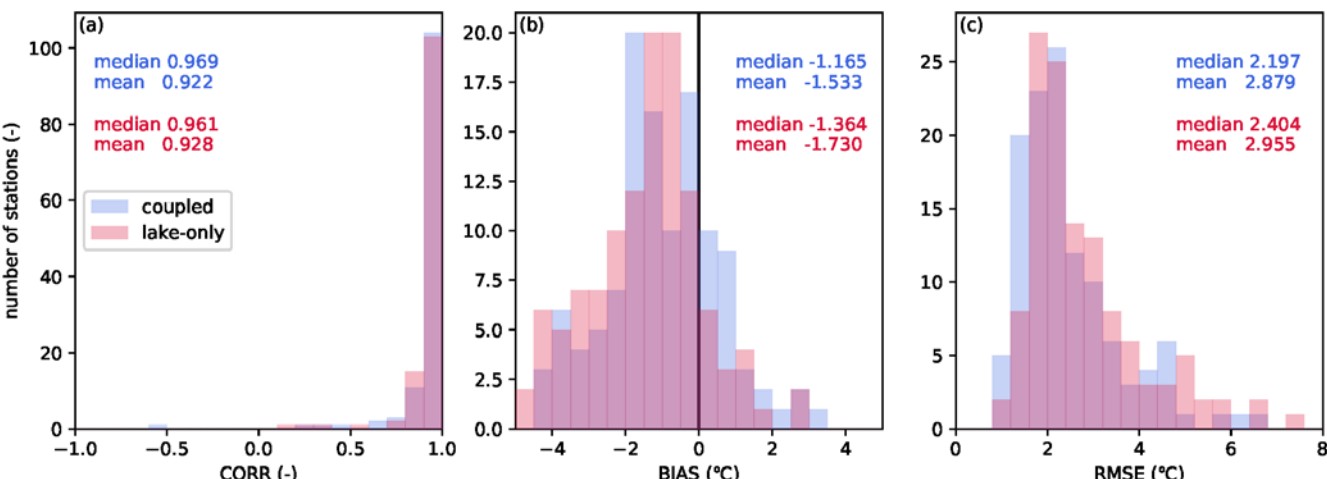

**Figure 10: Comparison of reproducibility indices for lake surface temperature. Bars indicate the histogram of each index, and characters do the media and mean value of them. Blue (red) bars and characters indicate the result of "coupled" ("lake-only") simulation. (a) CORR (-), (b) BIAS (°C), and (c) RMSD (°C).**

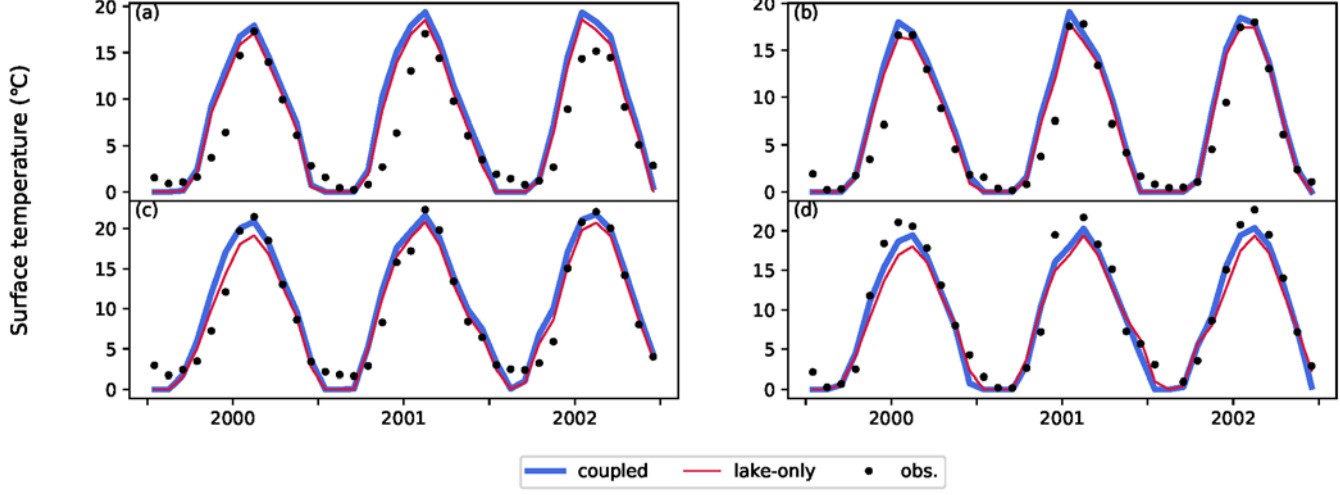

**Figure 11: Timeseries of monthly mean values of river temperature (°C). Black dots show the observed values, and blue (red) line shows the simulated values by "coupled" ("lake-only") simulation. (a) Lake Superior, (b) Ladoga, (c) Ontario, and (d) Champlain (the HydroLAKES ID is 5, 10, 7, and 64, respectively).**

## 5.7 Vertical profile of lake water temperature

Eleven lakes in North America were observed at multiple depths across the calculation period. In some cases, the simulated depth tended to be shallower than the maximum observed water depth because the lake bottom elevation parameter was derived from the mean lake depth of HydroLAKES. In general, the water surface elevation bias is considerably smaller than





this difference (according to the validation in Sect. 5.5). This bias exists due to the fact that the bathymetry of lakes is not spatially uniform. This underestimation of water depth can cause bias in water temperature (Stepanenko et al., 2013). Such a calibration was not conducted here; however, the model configuration fixed the bottom elevation derived from the datasets. Figure 12 shows three representative examples of vertical water temperature profile comparisons over six days.


As shown in the results for Lake Ontario and Huron in Figure 12 (a) and (b), the observed water depths in all the Great Lakes (except Lake Erie) are approximately double those of the simulated water depths. However, the vertical pattern of water temperature in the upper layers (up to approximately 60 m from the surface) in summer was reproduced well in all lakes. The "lake-only" simulation also reproduced the profile, but it was found that consideration of riverine in- and outflow

reduced the underestimation of surface temperature, which is in accordance with Sect. 5.6. However, the model still showed temperature overestimation in early spring (all four lakes were observed in April). The water temperature near the bottom was close to 4 °C, which is consistent with the maximum water density assumption at 4 °C. Moreover, this observation indicates a slightly lower temperature (2–3 °C) that we speculate may be a result of the entrainment of upper cooler water due to local mixing.




**Figure 12: Comparison between vertical water temperature profiles of simulations and observation: (a) Lake Ontario, (b) Lake Huron, and (c) Lake Oahe.**



Even for lakes smaller than the Great Lakes, the models tended to underestimate the temperature near the surface. Figure 12
(c) shows the results for Lake Oahe, where both experiments were shown to have underestimated the temperature near the
surface (up to approximately 20 m) in summer, regardless of the difference between calculated and observed water depths.
These trends were also observed in the satellite products described in Sect. 5.6.

## 6 Global distributions of lake impacts on riverine thermodynamics

The impact of reservoir operations and lakes affect downstream flow regimes (Hanasaki et al., 2006; Veldkamp et al., 2017).
Figure 13 shows the impacts of lakes on the global distribution of temperature changes in rivers. In most areas, the effect on
average temperature is within 1 °C. The inclusion of lakes lowers the river temperatures at high latitudes and in the Nile
River Basin and raises them in other regions (e.g., Parana River). It was found that the minimum river water temperature in
mid-latitudes during the cold season increased with inclusion of lakes (i.e., "coupled"). This is because the formation of
thermal stratification in the lakes warms the water near the surface. The opposite trend is observed in the Nile River, where
increase in heat loss (such as evaporation due to increased residence time) is dominant.







**Figure 13: Global distribution of the effect of introducing lakes on river water temperature, calculated by subtracting the simulated value of the "river-only" experiment that does not consider lakes from the "coupled" experiment that does (°C). Red color indicates that lakes cause an increase in river water temperature. (a) Annual mean of climatological monthly temperature, (b) minimum, and (c) maximum.**





At high latitudes, the minimum water temperature is at the freezing point and maximum water temperature decreases (up to 3 °C) in many basins. As Arctic rivers flow across a strong meridional temperature gradient, they play a role in transporting warmer water from upstream areas to colder downstream areas. However, the vertical one-dimensional lake model does not

correctly represent such an effect as the sub-lake-scale dynamics within the lake is not resolved. This impact becomes substantial for larger lakes that span multiple model grids.

## 7 Sensitivity to meteorological forcing dataset

We examined the sensitivity of the results to the forcing dataset by comparison with simulations using JRA55-ELSE (Kim, 2020) and Prcp-GPCCLW90 (Kim et al., 2009) forcing datasets in addition to the experiment based on the GSWP3 data.

Similar to Sect. 5, Figure 14 compares the performances of "coupled" and uncoupled ("river-only" and "lake-only" for riverine and lacustrine variables, respectively) forced by those three datasets. All simulations used the same river or river–lake network dataset. The stations subjected to validation are identical to those in Sect. 5.3–5.6.



Figure 14: Sensitivity of the performance metrics to the meteorological forcing datasets. Horizontal and vertical axes indicate the results of "coupled" and upcoupled ("river-only" for (a) and (b), "lake-only" for (c) and (d)) simulation, respectively. Gray, green, and brown color indicates the results of GSWP3, JRA55-ELSE, and Prcp-GPCCLW90, respectively. (a) River discharge, (b) river water temperature, (c) lake surface elevation, and (d) lake surface temperature. (i) CORR, (ii) BIAS, and (iii) RMSD. BIAS and RMSD are normalized for river discharge. BIAS and RMSD of the lake surface elevation in the "lake-only" simulation are not shown due to the drift even after spin-up.





In general, the results from the sensitivity experiments were similar to the GSWP3 results. The CORR and BIAS for river discharge were ameliorated by inclusion of lakes. The improved reproducibility in river water temperature was found in the "coupled" mode for all forcing datasets. Stable reproducibility of the lake surface elevation was found to be robust for the forcing datasets. Although incorporating the river model improved the underestimation of the lake surface temperature, a systematic bias is observed for lakes where the model overestimates the temperature. These under- and overestimation patterns can be attributed to the difference in heat capacity of river and lake waters. Shallow water depth in rivers leads to warmer temperatures due to more effective absorption of shortwave radiation. A better representation of vertical mixing may reduce such underestimation, which can lead to further realistic heat exchanges between the atmosphere.

## 8. Discussion for further development

The results discussed so far depend on the implementation of the models used within the integrated framework. While the river model employed in our study is a state-of-the-art model that can be applied on a global scale, it is evident that the lake model requires more improvement compared with previous studies (e.g., representation of the heat budget of bottom sediments and eddy mixing). This section summarizes potential improvements, mainly for the lake model.

The fundamental idea of the coupling framework is to represent only larger lakes in the river–lake network, with the aim of explicitly representing mass and energy exchanges with rivers upstream and downstream. Our study applied a one-dimensional vertical model to the larger lakes, but did not implement a sub-lake-scale model for the other lakes. However, it would be preferable if such a one-dimensional model is applied to smaller lakes from the perspective of the spatial heterogeneity of the actual lakes. Although the river–lake network developed in this study identifies the locations of lake inlets and outlet, the in-lake horizontal hydrodynamics are driven not only by river inflow and outflow but also by the uneven distribution of wind directions caused by surrounding topography and temperature gradients related to spatial heterogeneity of the bottom elevation. Such horizontal mixing could be one reason why there are changes in the optimal parameters or calculation schemes for vertical mixing depending on lake size (Subin et al., 2012). Previous studies have adopted an approach that divides lakes into horizontal grids and applies a vertical one-dimensional model to each column. This does not represent water flow in the horizontal direction. The computational cost of a three-dimensional model (Hodges, 2000) is very expensive to be applied globally, and hence, devising a simplified physical scheme is suggested. Such a model would contribute to our knowledge of the impacts of rivers on the thermodynamics of lakes, particularly in Arctic regions where incorporating vertical one-dimensional models results an underestimation of river water temperature.

This study also assumes that all outflows from lakes come from layers near the water surface. However, to minimize the impact of new dam construction on the ecosystem, some dams are manipulated to release water from a depth with the same



temperature as the water entering the reservoir. Furthermore, as highlighted in Sect. 4, while validating the upstream areas of reservoirs, the water balance is affected within and between basins as dam operations are conducted and conduits are built
between reservoirs. The latest lake dataset referred to in this study provides detailed information on the spatial distribution of lakes; however, information on outlets from lakes and reservoirs (e.g., location and height) also needs to be extended. Because the outlets of lakes coincide with the most downstream points of each unit-catchment grid in the river–lake network developed in this study, it will be easy to couple with dam operation models.

## 9. Conclusions

Our study was conducted to develop a coupling framework between a river model representing the horizontal flow and a lake model representing a layered structure based on locating lakes on a river network dataset to express the terrestrial hydrological transport of water and energy within the river–lake system. Two high-resolution datasets, MERIT Hydro and HydroLAKES, were merged and upscaled into a hydrological unit-catchment grid system. In our dataset, the upstream area of the reservoir was shown to correlate with the reported values well.


In-situ observation data on river discharge and water temperature and satellite datasets on water surface elevation and surface temperature of lakes were used to validate the framework. The global results show that the "coupled" simulation reproduced the absolute values and seasonal variations of those variables better than the individual river model or lake model. The effects of lakes on rivers and vice versa were then discussed by comparing simulation results, which showed better
representation in river discharge seasonality when lakes were introduced in the experiment. At sites where lakes occupy a large fraction in their associated basins, such as the Great Lakes region, an increase in evaporative loss from the lakes tended to improve the overestimation of absolute river discharge. It suggests that simultaneous representations of fluvial- and thermodynamics in rivers and lakes are necessary for reliable water availability estimates due to dam construction (Shiklomanov, 2000). The impact of coupling river and lake models on river water temperature has two main aspects: 1)
alleviating underestimations in mid-latitudes due to the formation of thermal stratification in lakes, and 2) causing a negative bias at high latitudes because of missing representation of sub-lake-scale dynamics within a lake suppressing poleward heat transportation by Arctic rivers.

Additionally, the contribution of river inflow and outflow to the water balance of lakes was significant. The seasonal
variation of water surface elevation was well reproduced on a global scale. The coupling effects on water surface temperature were not apparent. However, notably, the simulation and validation in this study did not consider spatial variability within lakes, which should be significant in larger lakes. The local energy budget of rivers and lakes is affected by water depth (Stepanenko et al., 2013) and water surface areas (Tokuda et al., 2019). The energy exchanges among them are determined by the combined impacts of their fluvial and thermodynamics. The impact of the water volume changes of lakes



and reservoirs has still not been elucidated, even in the latest global study (Vanderkelen et al., 2020), and the modeling
framework newly developed in this study, T-CHOIR, is expected to estimate a further reliable terrestrial heat budget.

**Code and data availability**

The source code of T-CHOIR can be obtained from https://zenodo.org/record/4584226. Due to the restrictions of the MERIT
Hydro data, the river-lake network data developed in this study is not available to public.

**Author contribution**

DT developed the river–lake network dataset and the coupling framework, analyzed the results. DT and HK developed the
manuscript. DY provided the topographic dataset and the original FLOW code. TO performed funding acquisition and
provided supervision. All authors contributed to discussions and improvement of the manuscript.

**Competing interests**

The authors declare that they have no conflict of interest.

**Acknowledgements**

This work was supported by the Japan Society for Promotion of Science (JSPS) via 16H06291. We gratefully acknowledge
the United Nations Environment Programme Global Environment Monitoring System (GEMS) for providing us with
monthly observed data of river water temperature, and the Global Runoff Data Centre (GRDC) for the daily observed
discharge data.

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
