# Peer review of "Development of a coupled simulation framework representing the lake and river continuum of mass and energy (TCHOIR v1.0)"

_Geoscientific Model Development, 2021_

## Author Comment (AC1)

**RC1: 'Comment on gmd-2021-75', Shuqi Lin, 13 Apr 2021**

**General comments:**

This study involves tremendous data processing when coupling the river and lake dataset together before conducting the simulations. Could you provide a map or a chart to state the number of river and lakes in different groups you defined and how many systems among them have been processed specifically? I think it will be helpful for the readers to understand the whole dataset and reproduce the framework.

> **Response:** Dear Dr. Lin, thank you for taking the time to review our manuscript. We are pleased your review is supportive of our work and we are happy to respond to the points. We added a table of pre-processing to enhance readers' understanding and uploaded the upscaled river–lake network dataset into zenodo. We also conducted additional validation with the data you recommended and it was really helpful for us to discuss the model applicability and limitation.

**Specific comments:**

**2.1 Harmonization of geographical information**

Did you basically implement the lake data from HydroLAKES and river data from MERIT Hydro?

I see a lot of preprocessing of lake and river geographical information in the second paragraph. Could you please provide a table or a chart to conclude the results of the preprocessing, like how many lakes are classified into the two groups, respectively, and how many inconsistencies are detected in two datasets and which dataset contained the largest upstream area you chosen in the end, etc.

> **Response:** Yes, the lake and river data were upscaled from HydroLAKES and MERIT Hydro, respectively. To show the variables the pre-processing updated, we added a table to summarize the pre-process as you recommended, which also indicates the manual processing.
>
> In addition, we added the number (369) of lakes resolved in the river–lake network dataset.
>
> We detected the inconsistency between the flow direction of MERIT Hydro and lake distribution of HydroLAKES and filled the gap in the pre-processing. However, those methods are minimal from the perspective of conduct of river–lake coupling simulation. We added a discussion to emphasize the importance of the development of a comprehensive geographical dataset explicitly representing rivers and lakes in the discussion section.

**Table 1 Summary of harmonization of geographical information, MERIT Hydro and HydroLAKES. All processes except number 4 are automated.**

| No. | Process | Updated variable | | Reference data |
|---|---|---|---|---|
| | | MERIT Hydro | HydroLAKES | |
| 1 | Select lakes to resolve in the river–lake network | - | lake area | lake area |
| 2 | Fill in isolated parts of each lake | - | lake area | lake area |
| 3 | Select lake outlet | - | - | upstream area calculated in MERIT Hydro |
| 4 | Remove outlets from endorheic lakes | flow direction | - | actual geography |
| 5 | Change flow direction in each lake | flow direction | - | lake outlet location |
| 6 | Recalculate upstream area for all the grids | upstream area | - | flow direction |

30    Line 89: Could you provide the links of these dataset here?

**Response:** As you recommended, we uploaded the dataset into the zenodo and updated the doi for the new version (the source code was not updated).

**3.2 Lake model**

Line 166: Any reference of this 1D lake model?

35    **Response:** The lake model in this study is not original because it is just a combination of each scheme of the existing models, so we only give an overview and cite corresponding studies.

Line 265: Should have a punctuation after the back bracket.

**Response:** Thank you for your correction. We added punctuation.

Line 298: How is the shortwave radiation weighted by the area of ice? Could you provide the equation here?

40    **Response:** We assume that incoming shortwave radiation and surface heat fluxes are boundary conditions for the energy budget of the water body, and they are different between ice-free and ice-covered lake areas. Shortwave radiation attenuates due to ice thickness in ice-covered areas. The heat flux into the water body is conductive heat beneath ice cover, or heat fluxes from and out of the atmosphere for the ice-free part. So, the model calculates the weighted mean of them with the ice-covered and ice-free areas. We added the explanation instead of equations.

 **3.3 Implementation of coupling interface**

Line 323: For how many river-lake systems in your study you have made the corrections? Are they the minor part of the whole dataset?
Why don't you leave off these particular systems to avoid the inaccuracy brought by the corrections?

> **Response:** "Correction" here is not related to the dataset, but it limits the river discharge and lake outflow not to exceed
50 > the corresponding storage at the previous timestep as the original river model, CaMa-Flood. I'm sorry for the ambiguous explanation and corrected the description.

**4 Validation of harmonized geographical information**

Table 1: Could you indicate these eight reservoirs in Fig 3 by different colors?

> **Response:** I updated Fig 3 with blue and red colors as you recommended:

[Figure]

55

**5.1 Simulation configuration**

Line 384 - 385: Could you mention this information at the beginning of the paper (maybe in section 2?)

> **Response:** We added the number of the lakes resolved into Sect. 2.1. (The information on volume and area are described only here because Sect. 2.1 focuses on the pre-processing.)

60 Line 407: Where are these initial values from?

**Response:** The "initial guess" of the initial value for lake water level and temperature were given with the surface elevation of lake water registered in HydroLAKES (attribute name is Elevation) and the air temperature on the first day of the calculation period, respectively. If the preliminary results with those values showed long-term drift, we manually set the new initial value after trial-and-error. In addition, the initial state of the vertical layer thickness was calculated from those values of lake water level, and the initial ice volume was set to be zero. We added the description and separate the paragraph into two; one for the initial condition and the other for model configuration.

**5.2 Reference data**

Line 444: You can get more vertical observations via https://www.glos.us/

**Response:** Thank you for letting us know about the portal site of precious observation data. We manually downloaded the part of the data, but it was difficult to proceed with the discussion on the validation because the model does not resolve the spatial variability of water temperature. We would like to utilize the data when we couple (simplified) 3D hydro- and thermodynamics lake models in further study, and added the discussion in the Sect. 7 (Discussion for further development).

**5.3 River discharge at downstream areas of lakes**

Figure 4: Could you please adjust the y-axis of (a) to integers?

**Response:** We updated the y-axis of histograms to integers in Fig. 4 and 10 as you recommended.

**5.5 Lake water surface elevation**

Figure 9: It looks like the "lake-only" simulation simulated much higher water surface elevation than the reality and the "coupled" simulation. Are these results from 20-year spin-up time run? In line 503, you mentioned that is due to the imbalance between precipitation and evaporation. Because I see the increase rate of the elevation in the "lake-only" simulation was not quite sharp. Can you initiate the model with the observations and try the simulation with less spin-up time?

If the imbalance between precipitation and evaporation could induce such a big discrepancy, the upstream rivers must have a big backflows when you change to the "coupled" simulation.

**Response:** Yes, these results for the "lake-only" experiment are also the results after 20-year spinning up. "Lake-only" experiment solves only the vertical mass budget, i.e., precipitation – evaporation. Because the surface temperature of lake water and evaporative flux is mainly governed by atmospheric conditions, the vertical mass budget does not reach

zero even after a very long spin-up period or any initial condition. The vertical mass budget is solved on a sub-daily scale, so the time series of water level has some seasonality and is not quite sharp, but the water level does not reach equilibrium range due to the abovementioned reason.

90

On the other hand, the model also considers riverine inflows and outflows, so a higher water level in lakes is consumed by the increase in outflows (increase in inflows from rivers to lakes could occur as you pointed out, but the increase in outflows is dominant). Consequently, the water level does not get too high but reaches the equilibrium range in the "coupled" simulation.

**5.7 Vertical profile of lake water temperature**

95

Line 598: Can you manually correct the lake depth? Because, especially in the Great Lakes, the incorrect lake depth may induce a big error in the thermal structure.

**Response:** Yes, we can correct the lake depth by editing a configuration file. We agree that the simulated thermal regime is sensitive to the lake depth. However, in the field of vertical 1D lake temperature model, there is still no consensus on how to set an appropriate lake depth (e.g. mean depth or the maximum depth) according to a model intercomparison project (Stepanenko et al., 2013). In this study, we consistently used the mean depth (attribute name is Depth_avg) contained in HydroLAKES dataset for all the lakes.

100

Line 582: Have you validated the ice simulation in the Great Lakes during early spring? The assumption of ice thickness in this model may affect the temperature simulation in the Great Lakes.

**Response:** As you recommended, we newly validated (1) ice cover period in each year and (2) monthly maximum of ice cover fraction in the Great Lakes and Lake St. Clair with dataset provided by GLERL (Assel, 2003; Wang et al., 2012). The model underestimates the period for all the lakes except for Lake St. Clair, which suggests that the vertical 1D lake model does not resolve the spatial distribution of the cover ice. Tuning of the ice shape parameterization could improve the bias as you discussed, and we think the implementation of a horizontal 2D (3D including vertical 1D) model is also a solution. On the other hand, we found that incorporation of river flows improved the underestimation of the monthly maximum of ice cover fraction in Lake Erie and St. Clair. This result suggests that cooler riverine inflow from the Northern area has an impact on the ice formation, and we confirmed that it has a similar effect on lake surface temperature. So, in Fig. 11 (d) showing the time series of surface water temperature in Lake Champlain, we replaced it with that in Lake St. Clair. We added those discussions into Appendix A1.

105

110

[Figure]

115

**Figure 11 (d) is updated from Lake Champlain to Lake St. Clair.**

[Figure]

120

**Figure A1: The comparison of (a) ice cover period in each year (day) and (b)-(g) monthly maximum of cover-ice fraction (%) in the Great Lakes region between the simulations and reference dataset. (a) The colored (white) face shows the results of the "coupled" ("lake-only") simulation. (b)-(g) Black dots show observed values, and the blue (red) line shows value simulated by the "coupled" ("lake-only") simulation. (b) Lake Superior, (c) Michigan, (d) Ontario, (e) Huron, (f) Erie, and (g) St. Clair (the HydroLAKES ID is 5, 6, 7, 8, 9, and 66, respectively).**

**7 sensitivity to meteorological forcing dataset**

125 Is this section necessary in the main body of this manuscript if the different meteo forcing did not generate obvious discrepancy?

**Response:** We move the section to Appendix A2 as you recommended.

**9 Conclusion**

Line 672: Please list some metrics here to show how much the "coupled" simulation is better than "river-only" and "lake-
130 only" simulations.

**Response:** We summarize the reproducibility indices in a table as you recommended.

**Table 2 Summary of comparison of reproducibility indices between coupled and uncoupled ("river-only" for riverine and "lake-only" for lacustrine variables) simulations.**

| Variable | Statistical Index (unit) | Coupled | Uncoupled ("river-only" or "lake-only") |
|---|---|---|---|
| River discharge | CORR (-) | 0.562 (0.462) | 0.482 (0.440) |
| | pBIAS (-) | -0.094 (0.106) | -0.080 (0.146) |
| | pRMSE (-) | 1.009 (1.276) | 1.093 (1.387) |
| River water temperature | CORR (-) | 0.894 (0.808) | 0.871 (0.807) |
| | BIAS (°C) | 0.723 (0.753) | 1.067 (0.914) |
| | RMSE (°C) | 2.493 (2.792) | 2.478 (2.814) |
| Lake water surface elevation | CORR (-) | 3.314 (0.274) | 0.343 (0.289) |
| | BIAS (m) | -1.594 (-0.818) | - (-) |
| | RMSE (m) | 3.479 (6.864) | - (-) |
| Lake surface temperature | CORR | 0.969 (0.922) | 0.961 (0.928) |
| | BIAS (°C) | -1.165 (-1.533) | -1.364 (-1.730) |
| | RMSE (°C) | 2.197 (2.879) | 2.404 (2.955) |

**Technical corrections:**

135 Figure 9: the unit of lake surface elevation should be (m) in the caption.

**Response:** Thank you for your kind correction. We corrected the unit in the caption.

---

## Author Comment (AC2)

**RC2: 'Comment on gmd-2021-75', Anonymous Referee #2, 20 Apr 2021**

The authors developed T-CHOIR that freely adjusts the spatial resolution of river-lake model that explicitly represents the energy and water balances in global scale. To achieve the objective, an improved flow upscaling algorithm, a hydrography dataset, and lake-reservoir dataset are tightly coupled. The authors identified and addressed many issues, which will help not only future users of the model but also general audiences working on the model and dataset developments. The manuscript was very well written, so it was great pleasure to read the manuscript. I only have several questions.

> **Response:** We appreciate your encouragement to improve our paper, and we want to reflect on all your suggestions. In particular, your comment on our dataset is highly critical and we reconfirmed the dataset in detail.

Line 63: What is the basis of saying "lower" and "higher"? These terms are comparative, but it is not straightforward to infer the comparisons. It would be also nice to briefly mention the reasons of lower water volumes and higher temperature.

> **Response:** We corrected the ambiguous explanation; Vanderkelen et al. (2020) concluded that the heat capacity of rivers has been decreasing due to a decrease in water volume, and that of lakes has been increasing due to a warm-up of water temperature. However, their model does not represent the temporal change in water volume in lakes.

Section 2.1: In case of lakes in a very upstream region, it is commonly found that the water body data of HydroLAKES lies between two basins of MERIT Hydro that drain to very different downstream. It is a universal problem that can exist in any DEM-derived flow direction dataset. How the T-CHOIR deal with this case? Do you correct the flow directions as done in MERIT-Hydro?

> **Response:** Thank you for your raise of an issue with the dataset. We would like to answer your comment from two perspectives.

> 1) Technical implementation. We modified the flow direction to reach a selected outlet for all the grids in each lake, so all the grids in each lake belong to the same basin. This modification changes basin size from a river-only upscaled map if a lake lies between two basins.

> 2) Actual situation for 369 lakes resolved in 15' resolution. We compared the spatial distribution of the lakes and that of basins in a river-only map upscaled only from MERIT Hydro, and it was found that 20 lakes lie between multiple basins. It was reasonable that six out of them are inland lakes (lakes without an outlet, e.g., the Caspian Sea and Lake Chad), and the river–lake network dataset deals with them as one basin. For most (13) of the other lakes, they are allocated the basin which has the most grids in each lake in the river-only map. Finally, there is only one exception,

Laguna Salada (HydroLAKES ID is 834). It is connected to the Colorado River basin, but the river occupies only 0.6% of the lake on the river-only map, but fortunately the river–lake network dataset reproduces the connection. As a result, we did not modify the dataset anymore from the first manuscript.

30

We added the above discussions to the manuscript.

Line 223: A lake may have multiple inflow paths. Does the model remember and update those inflows at every time step to calculate the "20% of inflow to lake"?

**Response:** Yes, we updated at every CFL timestep the sum of the inflow from all the inlets for each lake, then calculated the environmental flow with the total inflow at the previous timestep. We added the description there.

35

---

## Referee Report (RR1)

**General comments:**

In the revised manuscript, authors have addressed most of the questions I raised in the previous round. I appreciate the efforts authors made to improve the manuscript.

However, I think some minor revisions are required for publishing this study.

**Specific comments:**

**2.1 Harmonization of geographical information**

**Table 1:** Could you remove the first column of this table and add one more column to indicate the number of water systems (or the percentages of area and volume accounted for the total area and volume of all lakes) were processed by the specific method? I noticed that you add the information in the text, but it will be more intuitive to see them in the table.

**3.2 Lake model**

The reference for lake model mainly states where the input data come from. Could you refer some literature applying this model?

**5.7 Vertical profile of lake water temperature**

Figure 12: Why Lake Huron has two sets of observations?

Line: 601-602: Followed by my comment in the previous round, could you make a more specific discussion about the effect of underestimation of water depth, not just say it can cause bias? For example, the overestimation of water temperature in Lake Ontario and Huron (Figure 12) during the spring and fall time may be due to the underestimation of the water volume? And the coupled model, to some extent, enhanced the bias. Could you discuss the reason of these biases?

**Code and data availability:** authors have updated the source code and dataset in Zenodo, but to improve the reproducibility of the model, I think it is better to add a simple 'readme' file to clarify how to execute the model and what are the files in the 'map_glob_15min_lake' folder.

**Technical corrections:**

Line 674-678: Why is this sentence in bold?

Table 4: In the third column, what do the values inside the basket indicate? Could you add the explanation into the caption?

---

## Author Response (AR2)

Dear Dr. Lin, thank you for taking additional precious time to review our manuscript, and your comments are really supportive of improving our work. We are glad to answer each of your comments in the following. In the track-changes file, the updated parts are shown in blue color.

**Specific comments:**

5 **2.1 Harmonization of geographical information**

Table 1: Could you remove the first column of this table and add one more column to indicate the number of water systems (or the percentages of area and volume accounted for the total area and volume of all lakes) were processed by the specific method? I noticed that you add the information in the text, but it will be more intuitive to see them in the table.

**Response:** We add Table 3 in the Sect. 5.1 instead of this section as follows, because the resolved lakes depend on the

10 spatial resolution in an experiment. In addition, we apologize that we don't remove the first column because it is one of data processing.

**Table 1: Summary of the lakes resolved in the dataset and all the lakes in HydroLAKES (Values in the brackets indicate the fraction of the resolved lakes to all ones).**

| Type | Volume ($10^3$ km$^3$) | Area ($10^6$ km$^2$) |
|---|---|---|
| Lakes resolved in the 15-min dataset | 172.9 (92.0%) | 1.494 (51.1%) |
| All lakes in HydroLAKES | 187.9 | 2.927 |

**3.2 Lake model**

15 The reference for lake model mainly states where the input data come from. Could you refer some literature applying this model?

**Response:** The lake model was firstly developed for this manuscript, so we do not refer to any literature using it.

**5.7 Vertical profile of lake water temperature**

Figure 12: Why Lake Huron has two sets of observations?

20 **Response**: It is because observations were conducted at multiple sites on the same date.

Line: 601-602: Followed by my comment in the previous round, could you make a more specific discussion about the effect of underestimation of water depth, not just say it can cause bias? For example, the overestimation of water temperature in Lake Ontario and Huron (Figure 12) during the spring and fall time may be due to the underestimation of the water volume? And the coupled model, to some extent, enhanced the bias. Could you discuss the reason of these biases?

25 **Response:** We review the previous research again to update the discussion of the reproducibility of the water temperature profile in this study. It was reported that the bias (over- and underestimation) of water depth affects the

reproducibility of lake surface temperature via the heat capacity and vertical diffusion of the lake water, but the target site was a shallower lake (~5m) (Stepanenko et al., 2013).

However, according to Figure 12, our model overestimates the temperature during the winter season by about two Celsius degrees. It is thought that the model physics is the main reason for it. The model assumes that the maximum water density assumption at 4 °C and simulates the bottom temperature gets close to the temperature. On the other hand, the observed data shows a different pattern, in which temperature decreased to about 2 or 3 °C. The temperature difference implies that the conductive heat from the bottom sediment.

So we replace the discussions in the section as follows;

Figure 12 shows three representative examples of vertical water temperature profile comparisons over six days. As shown in the results for Lake Ontario and Huron in Figure 12 (a) and (b), in summer, the vertical water temperature pattern in the upper layers (up to approximately 60 m from the surface) was reproduced well in all lakes. The "lake-only" simulation also reproduced the profile, but it was found that consideration of riverine in- and outflow reduced the underestimation of surface temperature, which is in accordance with Sect. 5.6. The observed water depths in all the Great Lakes (except Lake Erie) are approximately double the simulated water depth. Previous research focusing on a much shallower lake reported that input water depth affects the reproducibility of the lake temperature via heat capacity and vertical diffusion (Stepanenko et al., 2013). Still, our results suggest that the energy exchange at the water surface is the governing factor in the season.

However, the model overestimated the temperature in early spring. The calculated water temperature near the bottom was close to 4 °C, consistent with the maximum water density assumption at four °C, while the observed data indicates a slightly lower temperature (2–3 °C). It is known that a more significant vertical mixing coefficient leads to a good reproducibility of the lake surface temperature (Gu et al., 2015), but it does not improve the overestimation in the entire depth in Lake Ontario. Therefore, the temperature gap between the observation and simulation can be attributed to the conductive heat from the bottom sediment requiring further studies to solve the bottom's energy budget.

**Code and data availability**

Authors have updated the source code and dataset in Zenodo, but to improve the reproducibility of the model, I think it is better to add a simple 'readme' file to clarify how to execute the model and what are the files in the 'map_glob_15min_lake' folder.

**Response:** We updated the zenodo repository to version 1.1 to add README for the map dataset as you suggested. So we also updated the zenodo URL to https://zenodo.org/record/5152668.

**Technical corrections:**

Line 674-678: Why is this sentence in bold?

60 **Response:** The manuscript was revised according to your instruction.

Table 4: In the third column, what do the values inside the basket indicate? Could you add the explanation into the caption?

**Response:** We revised the caption from "Table 4 Summary of comparison of reproducibility indices between coupled and uncoupled ("river-only" for riverine and "lake-only" for lacustrine variables) simulations" to "Table 4 Summary of reproducibility indices of coupled simulation. Values in the brackets are those of uncoupled ("river-only" for riverine 65 and "lake-only" for lacustrine variables) simulation."